# Harnessing the Role of Foliar Applied Salicylic Acid in Decreasing Chlorophyll Content to Reassess Photosystem II Photoprotection in Crop Plants

**DOI:** 10.3390/ijms23137038

**Published:** 2022-06-24

**Authors:** Michael Moustakas, Ilektra Sperdouli, Ioannis-Dimosthenis S. Adamakis, Julietta Moustaka, Sumrunaz İşgören, Begüm Şaş

**Affiliations:** 1Department of Botany, Aristotle University of Thessaloniki, 54124 Thessaloniki, Greece; moustaka@plen.ku.dk (J.M.); isgorensn@gmail.com (S.İ.); begum.sas99@gmail.com (B.Ş.); 2Institute of Plant Breeding and Genetic Resources, Hellenic Agricultural Organisation-Demeter (ELGO-Demeter), 57001 Thessaloniki, Greece; ilektras@bio.auth.gr; 3Section of Botany, Department of Biology, National and Kapodistrian University of Athens, 15784 Athens, Greece; iadamaki@biol.uoa.gr; 4Department of Plant and Environmental Sciences, University of Copenhagen, Thorvaldsensvej 40, 1871 Frederiksberg, Denmark

**Keywords:** chlorophyll fluorescence, light reactions, reactive oxygen species (ROS), photoinhibition, photodamage, non-photochemical quenching, oxidative stress, photosynthetic efficiency

## Abstract

Salicylic acid (SA), an essential plant hormone, has received much attention due to its role in modulating the adverse effects of biotic and abiotic stresses, acting as an antioxidant and plant growth regulator. However, its role in photosynthesis under non stress conditions is controversial. By chlorophyll fluorescence imaging analysis, we evaluated the consequences of foliar applied 1 mM SA on photosystem II (PSII) efficiency of tomato (*Solanum lycopersicum* L.) plants and estimated the reactive oxygen species (ROS) generation. Tomato leaves sprayed with 1 mM SA displayed lower chlorophyll content, but the absorbed light energy was preferentially converted into photochemical energy rather than dissipated as thermal energy by non-photochemical quenching (NPQ), indicating photoprotective effects provided by the foliar applied SA. This decreased NPQ, after 72 h treatment by 1 mM SA, resulted in an increased electron transport rate (ETR). The molecular mechanism by which the absorbed light energy was more efficiently directed to photochemistry in the SA treated leaves was the increased fraction of the open PSII reaction centers (q*p*), and the increased efficiency of open reaction centers (F*v*’/F*m*’). SA induced a decrease in chlorophyll content, resulting in a decrease in non-regulated energy dissipated in PSII (Φ*_NO_*) under high light (HL) treatment, suggesting a lower amount of triplet excited state chlorophyll (^3^Chl*) molecules available to produce singlet oxygen (^1^O_2_). Yet, the increased efficiency, compared to the control, of the oxygen evolving complex (OEC) on the donor side of PSII, associated with lower formation of hydrogen peroxide (H_2_O_2_), also contributed to less creation of ROS. We conclude that under non stress conditions, foliar applied SA decreased chlorophyll content and suppressed phototoxicity, offering PSII photoprotection; thus, it can be regarded as a mechanism that reduces photoinhibition and photodamage, improving PSII efficiency in crop plants.

## 1. Introduction

The world’s crop productivity is decaying while population growth is rising, placing increasing demands on agriculture [1]. The challenge of optimizing crop performance by increasing the efficiency and productivity of photosynthesis in crop plants is an essential and high priority research issue [1]. The increased efficiency and productivity of photosynthesis can be attained both by increasing the quantum yield of photosystem II (PSII) (Φ*_PSII_*) via decreasing photoprotective energy loss (NPQ) and through improving light energy distribution inside the canopy [1,2]. Decreasing leaf chlorophyll content was proposed as a promising approach to increase total canopy photosynthetic rate [2].

Salicylic acid (SA) is an essential plant hormone interconnected with other plant hormones such as auxin and plays a serious role in plant defense against pathogen infection [3]. Increased SA production and signaling during the activation of defense responses is related to a concomitant reduction in auxin biosynthesis and signaling, thus synchronizing defense and growth [3,4]. Salicylic acid belongs to the diverse group of plant phenolics which have been associated with the chemical defenses of plants against microbes, insects, and herbivores [4].

Exogenous application of SA has been reported to influence a range of developmental, physiological, and biochemical processes, e.g., seed germination, membrane permeability, stomatal closure and transpiration rate, growth, photosynthesis and yield [5]. However, it received much attention due to its role in modulating plant responses to biotic and abiotic stresses, acting as an antioxidant and a plant growth regulator [3,4,5,6,7]. Salicylic acid can mitigate the adverse effects of water deficit and salinity by ameliorating osmotic potential, membrane damage, stomatal conductance, transpiration rate, and biochemical parameters, restoring photosynthetic rates, and increasing shoot and root fresh and dry biomass, and also nutrient uptake [6,7]. Accumulation of SA under drought stress in *Avena sativa* influences stomatal opening, photorespiration and antioxidant defenses before any change in relative water content [8]. The beneficial role of 1 mM SA under salt stress in *Vigna angularis* is attributed to preventing the oxidative damage triggered by salinity stress [9]. Exogenous spray of SA can alleviate the adverse effects of Cd on growth and photosynthesis, promoting its antioxidant defense capacity [10]. Application of 10 µM SA significantly improved plant growth, the electron flow from Q_A_ to plastoquinone, and the efficiency of oxygen-evolving complex (OEC) in rice plants under Cd stress [11]. The beneficial role of SA under abiotic stress conditions is attributed to preventing the oxidative damage triggered by the stress conditions [9,10,11]. 

The photosynthetic apparatus of plants is subject to photoinhibition and, particularly, one of its major components, PSII [12]. Photoinhibition lowers the number of photochemically active PSII centers and is generalized across species, light conditions and habitats, lowering photosynthetic function and imposing substantial metabolic costs [13,14,15,16]. Chlorophyll fluorescence is widely used to measure photoinhibition, based on the ratio F*v*/F*m*, the maximum efficiency of PSII photochemistry [12,14].

The level of SA in Arabidopsis shoots extents to 1 µg^−1^ fresh weight, but it increases up to 20 µg g^−1^ at the place of pathogen attack [17,18]. SA accumulation stimulates pathogen-associated molecular patterns (PAMP), pathogen triggered immunity (PTI), effector-triggered immunity (ETI), and systemic acquired resistance (SAR) via the activation of plant defense genes to resist biotrophic pathogens [3,18,19]. The consequences of exogenously applied SA on plant physiological processes under optimal growth conditions are controversial, with reports having a positive effect on plant growth or others having a negative influence on various physiological processes [20]. Following spray application of SA in soybean and corn, photosynthetic rates increased, possibly as a result of increased enzyme activity related to CO_2_ uptake, rather than increases in stomatal opening [20,21]. Addition of 0.5 mM SA to hydroponic culture of maize plants decreased net photosynthesis under both low light (LL) and high light (HL) conditions, together with a decrease in the stomatal conductance (*g*_s_) and transpiration rate [22]. In contrast, the same concentration provided protection to low temperature-induced damage in young maize [22] or alleviated the damaging effect of paraquat in barley [23]. SA application may influence the process of photosynthetic electron transport via enhancing the non-photochemical fluorescence quenching (NPQ) mechanism [20,24,25]. However, the exact mechanism of SA action on the electron transport process is still unclear [20].

Measurements of chlorophyll *a* fluorescence have been extensively used to probe the functions of photosynthetic machinery, especially of PSII [15,26,27,28]. Chlorophyll *a* fluorescence measurements can provide information about the direct or indirect effects of SA on the light energy utilization in PSII [24]. The light energy absorbed by PSII antennae can be either utilized via photochemistry or competitively dissipated via various thermal processes [29,30,31,32]. The conversion efficiency of absorbed light energy to photochemical energy is critical in describing plant productivity over time [33]. The light reactions of photosynthesis involve a set of redox reactions that are the source of reducing power and energy for producing organic compounds [34]. When the light energy absorbed by photosystem II (PSII) and photosystem I (PSI) exceeds the amount that can be used for photochemistry, the consequence is the increased formation of reactive oxygen species (ROS), such as singlet oxygen (^1^O_2_), superoxide anion radical (O_2_**^•^**−), and hydrogen peroxide (H_2_O_2_) [35,36,37,38,39]. Under excess light conditions, the overexcitation of PSII increases the probability of the formation of the triplet chlorophyll state (^3^Chl*) from the singlet excited states (^1^Chl*) through the intersystem crossing, producing single oxygen (^1^O_2_) [39,40,41,42,43]. Electron leakage to O_2_ at PSI results in the superoxide anion radical (O_2_^•^−) that via a disproportionation reaction is catalyzed by superoxide dismutase (SOD) and is converted to hydrogen peroxide (H_2_O_2_) [44,45], which is the most stable and least reactive ROS with the longest lifetime, as it is able to easily diffuse through the membranes [46]. To prevent the formation of ROS and photoinhibition, the absorbed light energy by the light-harvesting complexes must match the rate of electron transport from PSII to PSI [47,48].

In the present study, we evaluated under optimal growth conditions of tomato plants the consequences of foliar applied 1 mM SA for 72 h on photosystem II (PSII) function and estimated the effects of SA application on reactive oxygen species (ROS) generation and chlorophyll content. Our hypothesis was that foliar application of SA will increase the quantum yield of photosystem II (PSII) (Φ*_PSII_*) via decreasing the photoprotective energy loss (NPQ). Thus, through improving light energy distribution, SA will optimize tomato crop performance by increasing the efficiency and productivity of photosynthesis. Consequently, foliar application of SA can be regarded as a mechanism that reduces photoinhibition and photodamage, improving PSII efficiency in crop plants.

## 2. Results

### 2.1. Chlorophyll Content and Maximum Efficiency of Photosystem II of Control and Salycilic Acid Treated Tomato Leaves

In each independent experiment of foliar sprayed tomato leaves by either distilled water (control) or 1 mM SA, chlorophyll contents were assessed after 72 h of treatment (Figure 1a). The chlorophyll content, expressed in relative units, decreased significantly after 72 h of treatment with 1 mM SA (Figure 1a). In contrast to this, the maximum efficiency of PSII photochemistry (F*v*/F*m*) of tomato leaves increased compared to the control after 72 h of SA treatment (Figure 1b).

### 2.2. Light Energy Utilization in Photosystem II of Control and Salycilic Acid Treated Tomato Leaves

The changes in the light energy utilization in PSII of foliar sprayed tomato leaves, by either distilled water (control) or 1 mM salicylic acid (SA), were estimated by measuring the amount of light energy allocated for photochemistry in PSII (ΦPSII) for regulated non-photochemical energy loss; that is, for photoprotective heat dissipation (ΦNPQ) and for non-regulated energy loss in PSII (ΦNO), the sum of them to be equal to 1 [49].

The effective quantum yield of PSII photochemistry (Φ*_PSII_*), after 72 h of treatment by 1 mM SA, increased by 11% at low light (LL) and 7.5% at high light (HL), compared to the controls (Figure 1c). At the same time, the quantum yield of regulated non-photochemical energy loss (Φ*_NPQ_*) decreased by 1 mM SA treatment at LL without any change at HL (Figure 1d), while the quantum yield of non-regulated energy (Φ*_NO_*) did not change by SA treatment at LL and decreased at HL compared to their controls (Figure 1e).

### 2.3. Changes in the Photoprotective Heat Dissipation, the Redox State of the Plastoquinone Pool and the Efficiency of Open Photosystem II Reaction Centers after Salycilic Acid Treatment

Non-photochemical quenching (NPQ), which reflects heat dissipation of excitation energy, decreased after 72 h of treatment by 1 mM SA at 205 μmol photons m^−2^ s^−1^ actinic light (AL) illumination (LL) compared to control plants (sprayed by distilled water), while there was no difference at 1000 μmol photons m^−2^ s^−1^ actinic light (AL) illumination (HL) compared to control plants (Figure 1f).

The redox state of quinone A (Q*_A_*), representing the fraction of open PSII reaction centers (q*p*), increased by 1 mM SA treatment at both LL and HL actinic light (AL) illuminations (Figure 2a). At the same time, salicylic acid (SA)-treated plants, showed a significant increase in the efficiency of excitation energy capture by open PSII centers (F*v*’/F*m*’), at both LL and HL actinic light (AL) illumination (Figure 2b).

### 2.4. Changes in the Electron Transport Rate and the Excitation Pressure after Salycilic Acid Treatment

The electron transport rate (ETR) increased by 1 mM SA treatment at both LL and HL (Figure 2c), with a parallel decrease in excitation pressure at PSII (1 − q*p*) at both actinic light (AL) illuminations (Figure 2d).

### 2.5. Changes in the Excess Excitation Energy in Photosystem II and the Efficiency of the Oxygen Evolving Complex after Salycilic Acid Treatment

The excess excitation energy (EXC), which is calculated as (F*v*/F*m* − Φ_PSII_)/F*v*/F*m*, decreased after 72 h of treatment by 1 mM SA at both actinic light (AL) illuminations (Figure 3a). The efficiency of the oxygen evolving complex on the donor side of PSII (F*v*/F*o*) in tomato leaves after 72 h of treatment by 1 mM SA increased compared to the control plants (sprayed by distilled water) (Figure 3b).

### 2.6. The Spatial Pattern of Photosystem II Activity of Control and Salycilic Acid Treated Tomato Leaves

The whole tomato leaf pattern after 72 h of treatment by 1 mM SA is shown in Figure 4, with the parameter of the minimum chlorophyll *a* fluorescence in the dark (F*o*) to decrease, while the maximum chlorophyll *a* fluorescence in the light (F*m*‘) to increase (Figure 4).

The maximum efficiency of PSII photochemistry (F*v*/F*m*) increased significantly after 72 h of treatment by 1 mM SA, while the non-photochemical quenching (NPQ), depicted as NPQ/4, decreased compared to tomato leaflets sprayed with distilled water (control) (Figure 4). The effective quantum yield of photochemistry (Φ*_PSII_*) increased significantly after 72 h of treatment by 1 mM SA, while the quantum yield of regulated non-photochemical energy loss (Φ*_NPQ_*) decreased to such an altitude that the quantum yield of non-regulated energy (Φ*_NO_*) remained constant compared to tomato leaflets sprayed with distilled water (control) (Figure 5). The fraction of open PSII reaction centers (q*p*) increased significantly after 72 h of treatment by 1 mM SA compared to the control (Figure 5).

### 2.7. Reactive Oxygen Species Imaging of Control and Salicylic Acid Treated Tomato Leaves

Reactive oxygen species imaging, which was performed with 25 µM 2’, 7’-dichlorofluorescein diacetates in the dark, revealed a decrease in ROS generation in 1 mM SA of tomato leaflets after 72 h of spray (Figure 6b) compared to tomato leaflets sprayed by distilled water (control) (Figure 6a). A higher ROS generation was visible in control leaves, as green fluorescence localized mainly in leaf veins and leaf hairs (Figure 6a).

## 3. Discussion

Salicylic acid (SA), introduced as the “sixth” principal phytohormone only in the early 1990s [19], is one of the key hormones of plant disease resistance [50,51] and can also directly or indirectly affect various physiological processes, including photosynthesis [20]. Experimental evidence showed that SA exerts inhibitory effects on chlorophyll content, stomatal function, photosynthetic parameters, and carboxylating enzymes, playing a regulatory role associated with photosynthetic reactions [52]. SA takes part in plant responses to many abiotic stresses such as drought, salinity, chilling, heat, and heavy metal toxicity [10,11,53,54,55,56,57,58]. However, since its mode of action depends significantly on numerous aspects, such as the plant species, the environmental conditions, the concentration used, the duration of exposure, and the concentrations of other external agents [4,20,59], data on exogenously applied SA on plant physiological processes under stress or non-stress conditions remain controversial [20].

The fundamental organelles in photosynthesis are the chloroplasts that are composed of thylakoids, in which reside the antenna of the light-harvesting complexes (LHCI, LHCII) that capture the light energy by the photosynthetic pigments, including chlorophylls [60]. Chlorophyll molecules are the main pigments in absorbing light quanta and transferring the energy to the reaction centers where charge separation happens and electron transport begins [40,61]. In case the absorbed light energy exceeds the capacity of photosynthesis to use it for assimilation, generation of high levels of ROS occur [61,62,63]. Although excess light is potentially detrimental, plants have developed several mechanisms to cope with the excess absorbed energy to prevent photooxidative stress [61,62,63]. Photooxidative stress results in photodamage, either at the acceptor side through triplet excited state chlorophylls (^3^Chl*), resulting in singlet oxygen (^1^O_2_) formation, or at the donor side through inactivation of the oxygen evolving complex (OEC) [64]. Plants possess limited capability to regulate the amount of absorbed sunlight through changes in leaf angle, leaf area, chloroplast movement, and, on a molecular level, through acclimatory adjustments of antenna size [2].

Plants with higher chlorophyll content have larger antenna size and can capture and absorb more light energy [65]. On the contrary, plants with reduced chlorophyll content have smaller chlorophyll antenna size, representing reduced NPQ and ROS generation [65]. Decreasing leaf chlorophyll concentration has been suggested as a potential method to reduce excess absorption of sunlight and improve photosynthetic efficiency [1,2,65,66,67]. Improved photosynthetic efficiency is achieved via a better allocation of absorbed light energy that also reduces photooxidative stress [68].

When mustard leaves were sprayed with 0.01 mM SA, the chlorophyll content was significantly enhanced, whereas higher SA concentrations lessened chlorophyll content [69,70]. In barley and black gram plants pre-treated with SA, a decrease in chlorophyll content was detected [71,72]. In our experiment, the higher chlorophyll content observed in control leaves (Figure 1a) was correlated with lower values of maximal fluorescence yield at the light-adapted state (F*m*’) (Figure 4), as has also been pointed out previously [73]. These lower values of F*m*’ were corelated with higher photoprotective energy dissipation as heat (NPQ) in control leaves compared to SA treated (Figure 4), indicating photoprotective effects provided by the foliar application of SA. Photoprotection by NPQ is important for optimal growth and development [74,75]. A lower value of minimal fluorescence yield at the dark-adapted state (F*o*), as we noticed in SA treated tomato leaves (Figure 4), was shown to indicate photoprotective effects provided by anthocyanins in the red leaf parts [73]. In accordance with this, tomato plants, grown in non-stressed conditions and water sprayed (control), reached an NPQ value of ~0.7 at 205 μmol photons m^−2^ s^−1^, while 1 mM SA sprayed reached a lower amplitude (~0.5) (Figure 1f). This indicates that in SA sprayed leaves there was lesser photoprotective need. Still, the decreased NPQ in SA sprayed leaves indicates that its action minimizes the potential for photooxidative damage. Additionally, the lower value of minimal fluorescence yield at the dark-adapted state (F*o*) in SA sprayed leaves has been shown to be correlated with photoprotection [73].

Salicylic acid was shown to slow down PSII electron transport rate (ETR) [76] but according to our results, tomato plants sprayed with SA showed an enhancement of ETR and increased effective quantum yield of photosystem II (PSII) photochemistry (Φ*_PSII_*) at both LL and HL (Figure 1c). The increased quantum yield of PSII photochemistry (Φ*_PSII_*) according to Genty et al. [77] can be attributed either to the increased fraction of open PSII reaction centers (q*p*), which is a measure of the redox state of quinone A (Q*_A_*), or to the increased efficiency of these centers (F*v*’/F*m*’). In our case, the increased Φ*_PSII_* was due to both the increased fraction of open PSII reaction centers (Figure 2a) and to the increased efficiency of these centers (Figure 2b). In SA sprayed leaves, a larger amount of absorbed light energy was distributed to PSII photochemistry, as revealed by the significant increase in Φ*_PSII_* (Figure 1c) and ETR (Figure 2c).

Salicylic acid (1 mM), supplied in the nutrient solution of tomato plants for 24 h, decreased the maximum efficiency of PSII (F*v*/F*m*), and the effective quantum yield of PSII (Φ*_PSII_*) [78], but in our experiment on SA sprayed leaves, we observed the reverse effects, with SA resulting in increased F*v*/F*m* and Φ*_PSII_* values (Figure 1b,c). Lower F*v*/F*m* values, as observed in distilled water (control) sprayed tomato leaves (Figure 1b), indicate a higher degree of photoinhibition [12,14]. It seems that most results obtained with exogenous SA cannot be generalized, since the effect may differ not only with plant species, but it may vary on the method of SA application (foliar spray, growth medium addition, seed pre-soaking, etc.), as well as on the experimental duration time [76].

Despite the lower chlorophyll content of SA-treated leaves (Figure 1a), the absorbed light energy was preferentially converted into photochemical energy (Figure 1c) rather than dissipated as thermal energy by NPQ (Figure 1f). The on-photochemical quenching (NPQ) mechanism can reduce the energy transfer to reaction centers [30] and represents mainly the thermal energy dissipation from LHCII via the zeaxanthin quencher [40,79,80]. The increased heat dissipation by de-excitation (NPQ) in control plants (Figure 1f) decreases the efficiency of photochemical reactions of photosynthesis (Figure 1c) (downregulation of PSII) [30,44,74,81,82,83]. Thus, the decreased non-photochemical quenching (NPQ) (Figure 1f) and relative excess energy at PSII (EXC) (Figure 3a) in SA treated leaves indicated improvements related to PSII efficiency. Photosystem II responses triggered by the NPQ mechanism is an approach to protect the photosynthetic apparatus from photo-oxidative damage by dissipating excess light energy such as heat at PSII [48,84,85,86], regulating photosynthetic ETR [87,88] and avoiding detrimental ROS creation [48,81]. The downregulation of PSII is a reversible decrease in the quantum yield of PSII under excess light conditions. Photoinhibition, also called photoinactivation, lowers the number of photochemically active PSII centers and recovery occurs only via degradation and resynthesis of the D1 protein of the photoinhibited PSII centers [16]. SA has been previously reported to act as a signal molecule to activate chlorophyll catabolic genes [32]. Our results point out that an SA stimulated decrease in chlorophyll content can be regarded as a mechanism that can lower photoinhibition and photodamage of PSII, while the blockage of a decrease in chlorophyll content intensifies PSII photoinhibition and photodamage [32].

Increasing concentrations of SA (0.1 mM, 0.5 mM and 1 mM), imposed through the root medium for a period of 7 day on barley (*Hordeum vulgare* L.), caused a concentration dependent inhibitory effect on PSII function, by inhibiting the electron donation from the oxygen evolving complex (OEC) to PSII reaction centers, as a result of some structural changes in the catalytic Mn cluster at protein level, resulting in a decrease in the fraction of open of PSII centers [52]. In contrast, in our experiment, a lower efficiency of the oxygen evolving complex (OEC) was noticed in the untreated (control) tomato leaves, while foliar application of SA resulted to enhanced efficiency of the OEC (Figure 3b) and to an increased fraction of open of PSII reaction centers (Figure 2a). Thus, in addition to the concentration of SA used; the choice of application method is critical for its impact on a plant’s physiology. The main site of NPQ is the marginal PSII antenna [86] but quenching by the reaction centers is also probable [89], especially under a low OEC efficiency [83], possibly explaining the increased NPQ at LL in the untreated (control) tomato leaves (Figure 1f). The decreased efficiency of the OEC is associated with the formation of H_2_O_2_, which can be further oxidized to the superoxide radical (O_2_^•^−) [90]. This could explain the increased ROS formation in control leaves compared to SA-treated leaves (Figure 6).

When the rate of excitation energy reaching PSII is higher than can be used in photochemistry, this leads to increased “excitation pressure” (1 − q*p*) on PSII, which can be measured non-invasively using the saturating light pulse method of chlorophyll fluorescence [90]. Foliar application of 1 mM SA on tomato leaves decreased the excitation pressure at PSII (1 − q*p*), compared to the untreated one (Figure 2d), also decreasing the excess excitation energy (EXC) (Figure 3a). The conversion efficiency of absorbed light energy to photochemical energy is critical in describing plant productivity over time [33].

When plants are under a HL environment, avoiding photosynthetic yield decline is considered critical [91]. Photosystem II over-excitation increases the conversion of unquenched singlet excited state chlorophylls (^1^Chl*) into triplet excited state chlorophylls (^3^Chl*), accompanied by ^1^O_2_ generation that damages and inhibits PSII [29,39,40,64,91,92,93]. The role of ^1^O_2_ is ambiguous; under moderate stress, it acts as a retrograde signal, whereas under severe stress, it causes oxidative damage [94]. Under HL, both control and SA treated plants showed an equal maximum value in the quantum efficiency of thermal dissipation (Φ*_NPQ_*) (Figure 1d), and a minimum value in Φ*_PSII_*, which was even lower in the control than SA-treated plants (Figure 1c). As a result, under HL, control plants displayed higher values of non-regulated energy dissipated in PSII (Φ*_NO_*) than SA-treated plants (Figure 1e). A decreased Φ*_NO_* at HL by the SA treatment compared to controls (Figure 1e) implies a better photoprotection and is indicative of lower ^1^O_2_ production [95]. The inability of PSII to utilize the absorbed light energy for photochemistry (Φ*_PSII_*), or to safely dissipate it as heat (Φ*_NPQ_*), can lead to the formation of a triplet chlorophyll state (^3^Chl*) that can react with O_2_ to produce the very reactive ^1^O_2_ [93,95]. The probability of formation of ^3^Chl* can be calculated by Φ*_NO_* [96]. An increased Φ*_NO_* reflects the inability of a plant to protect itself against damage by excess illumination, which will eventually lead to photodamage [97,98], indicating that the plant has problems coping with the incident HL [95].

Singlet oxygen (^1^O_2_) produced by the triplet excited state chlorophylls (^3^Chl*) can further produce the other ROS, e.g., O_2_^•^− and H_2_O_2_ [32,99]. However, electron leakage to O_2_ at PSI, resulting in O_2_^•^−, which is converted to H_2_O_2_, is the main pathway of H_2_O_2_ generation [39,44,45]. Increased ^1^O_2_ generation is considered to be harmful, since it can inhibit the repair of PSII reaction centers and/or contribute directly in oxidation and PSII damage, or even more activate programmed cell death [39,63,87,93,98,99]. Singlet oxygen production is considered to act as the agent of photo-inhibitory damage [16].

Among the ROS, ^1^O_2_ and H_2_O_2_ initiate diverse signaling networks when photosynthesis is disturbed [100]. Singlet oxygen, since it is very reactive, initiates but does not transduce signaling. On the other hand, H_2_O_2_, due to its lower reactivity, is a mobile messenger in a spatially defined signaling pathway [100]. Foliar produced chloroplastic H_2_O_2_ shows preferential accumulation in leaf bundle sheath cells [100,101,102,103] (Figure 6). Hydrogen peroxide can be transmitted throughout the leaf veins to act as a long-distance signaling molecule [99,103].

Tomato leaves sprayed with 1 mM SA with a lower chlorophyll content compared to water sprayed (Figure 1a) are characterized by decreased non-regulated energy dissipated in PSII (Φ*_NO_*) under high light treatment (Figure 1e), suggesting a lower amount of triplet excited state chlorophyll (^3^Chl*) molecules available to produce singlet oxygen (^1^O_2_). The decreased chlorophyll content in tomato plants stimulated by SA treatment can be regarded as a mechanism that can lower the photoinhibition and photodamage of PSII [1,2], and has been regarded as a way to suppress phototoxicity in plants [32,104]. Block of chlorophyll catabolism exacerbates PSII photoinhibition and photodamage [32]. The molecular mechanism by which, in SA-treated leaves with lower chlorophyll content, the absorbed light energy is more efficiently directed to photochemistry, was the increased efficiency of the reaction centers (F*v*’/F*m*’) (Figure 2b), and the increased fraction of open PSII reaction centers (q*p*) (Figure 2a), which enhanced ETR (Figure 2c).

Tomato leaves sprayed with 1 mM SA presented a decrease in chlorophyll content associated with higher PSII photochemistry (Φ*_PSII_*) and electron transport rate (ETR), but also with lower excitation pressure (1 − q*p*) and lower excess excitation energy (EXC). 1 mM SA sprayed tomato leaves also possessed a higher capacity to maintain low Φ*_NO_* under HL (Figure 1e), reflecting the ability of SA sprayed tomato leaves to be protected against damage from excess illumination.

The molecular mechanism by which SA can induce the decrease in chlorophyll content under optimum growth conditions may be a field worth further investigation. How this chlorophyll loss occurred may differ from the pathway of chlorophyll catabolism during leaf senescence in essential aspects of gene regulation and biochemistry.

## 4. Materials and Methods

### 4.1. Plant Material and Growth Conditions

Tomato (*Lycopersicon esculentum* Mill. cv. Meteor) plants were purchased from the market and transferred to a growth chamber with a 14 h photoperiod, 20 ± 1/19 ± 1 °C day/night temperature, relative humidity 50 ± 5/60 ± 5% day/night [103] and photosynthetic photon flux density (PPFD) 200 ± 10 μmol quanta m^−2^ s^−1^.

### 4.2. Salicylic Acid Treatment

Tomato (*Lycopersicon esculentum* Mill. cv. Meteor) plants were sprayed either with distilled water (control) or with 1 mM salicylic acid (SA). All treatments were performed with four independent biological replicates.

### 4.3. Chlorophyll Content

The relative chlorophyll content of the control (sprayed with distilled water) or 1 mM SA-treated leaves for 72 h was measured photometrically with a portable Chlorophyll Content Meter (Model Cl-01, Hansatech Instruments Ltd., Norfolk, UK), using dual wavelength optical absorbance (620 and 920 nm) and expressed in relative units [73].

### 4.4. Chlorophyll Fluorescence Analysis

Chlorophyll fluorescence measurements were performed in dark-adapted (20 min) tomato control and SA sprayed plants using an Imaging-PAM Fluorometer M-Series MINI-Version (Heinz Walz GmbH, Effeltrich, Germany). Chlorophyll fluorescence analysis was performed as described in detail previously [105]. Measurements were conducted on tomato leaves after 72 h of foliar spray, either by distilled water (control) or 1 mM SA. Six to nine areas of interest (AOI) were selected in each leaflet. For each AOI, we measured the minimum (F*o*) and the maximum (F*m*) chlorophyll *a* fluorescence in the dark. The maximum chlorophyll *a* fluorescence in the light (F*m*‘), with saturating pulses (SPs was computed every 20 s for 5 min after application of the actinic light (AL), while the minimum chlorophyll *a* fluorescence in the light (F*o*‘) was computed as F*o*‘ = F*o*/(F*v*/F*m* + F*o*/F*m*‘) [106], where F*v* is the variable chlorophyll *a* fluorescence (in the dark-adapted leaves), calculated as F*m* − F*o*. Steady-state photosynthesis (F*s*) was measured after 5 min of illumination time, either with AL of 205 μmol photons m^−2^ s^−1^ corresponding to low light intensity (LL), or with 1000 μmol photons m^−2^ s^−1^ corresponding to high light intensity (HL). By using Win software (Heinz Walz GmbH, Effeltrich, Germany), we estimated the maximum efficiency of PSII photochemistry (F*v*/F*m*), the effective quantum yield of PSII photochemistry (Φ*_PSII_*), the quantum yield of regulated non-photochemical energy loss (Φ*_NPQ_*), and the quantum yield of non-regulated energy (Φ*_NO_*), according to Krammer et al. [49]. The efficiency of the oxygen evolving complex (OEC) on the donor side of PSII (F*v*/F*o*) [30,48,107,108], the efficiency of excitation energy capture by open PSII centers (F*v*’/F*m*’) [77], the redox state of quinone A (Q*_A_*), representing the fraction of open PSII reaction centers q*p* = (F*m*‘− F*s*)/(F*m*‘ − F*o*‘) [77], and the non-photochemical quenching (NPQ), reflecting the dissipation of excitation energy as heat were calculated as (F*m* − F*m*‘)/F*m*‘ [109]. The electron transport rate (ETR), calculated as Φ_PSII_ × PAR × c × abs, where PAR is the photosynthetically active radiation, c is 0.5, and abs is the total light absorption of the leaf taken as 0.84 [110], the excitation pressure (1 − q*p*), and the excess excitation energy (EXC) as (F*v*/F*m* − Φ*_PSII_*)/(F*v*/F*m*) [111] were also calculated.

Representative results as color-coded images of the minimum chlorophyll *a* fluorescence in the dark (F*o*), the maximum chlorophyll *a* fluorescence in the light (F*m*‘), the maximum efficiency of PSII photochemistry (F*v*/F*m*), the non-photochemical quenching (NPQ), depicted as NPQ/4, the effective quantum yield of PSII photochemistry (Φ*_PSII_*), the quantum yield of regulated non-photochemical energy loss (Φ*_NPQ_*), the quantum yield of non-regulated energy (Φ*_NO_*), and the fraction of open PSII reaction centers (q*p*), are also shown.

### 4.5. Imaging of Reactive Oxygen Species

Imaging of ROS in tomato leaflets was performed after 72 h of foliar spray, either by distilled water (control) or 1 mM SA, as described previously [112]. ROS specific fluorescence in tomato leaflets were observed with a Zeiss AxioImager Z2 epi-fluorescence microscope, equipped with an AxioCam MRc5 digital camera [113]. Leaves were previously incubated with 25 μM 2′, 7′-dichlorofluorescein diacetate (DCF-DA, Sigma Aldrich, Chemie GmbH, Schnelldorf, Germany) for 30 min in the dark [114].

### 4.6. Statistics

Pairwise differences in the control and SA treatments were analyzed with independent samples *t*-test for unequal variances, using the IBM SPSS Statistics for Windows version 28 at a *p* < 0.05 level.

## 5. Conclusions

We conclude that foliar applied SA induced a decrease in chlorophyll content, lowering ROS generation and enhancing PSII efficiency. However, the impact of SA on plant growth and physiology relies highly on the study design strategy and depends significantly on numerous aspects, such as the plant species or the genotype, the environmental conditions, the concentration used, the duration of exposure, and, mainly, the method of application (foliar, rooting medium, e.g., soil or hydroponic). According to our experimental data, foliar application of 1 mM SA by spray, compared to other application methods, has the best beneficial effect, alleviating photoinhibition and photodamage and improving PSII efficiency in the crop plant, tomato.

This study sheds light on the role that chlorophyll molecules play in the regulation of PSII efficiency, as well as on ROS generation. The results extend our mechanistic understanding of SA-induced enhancement of PSII efficiency through a decreased chlorophyll content that suppressed phototoxicity, thus it can be regarded as a mechanism that lowers photoinhibition and photodamage of PSII in the important horticultural crop, tomato.

## Figures and Tables

**Figure 1 ijms-23-07038-f001:**
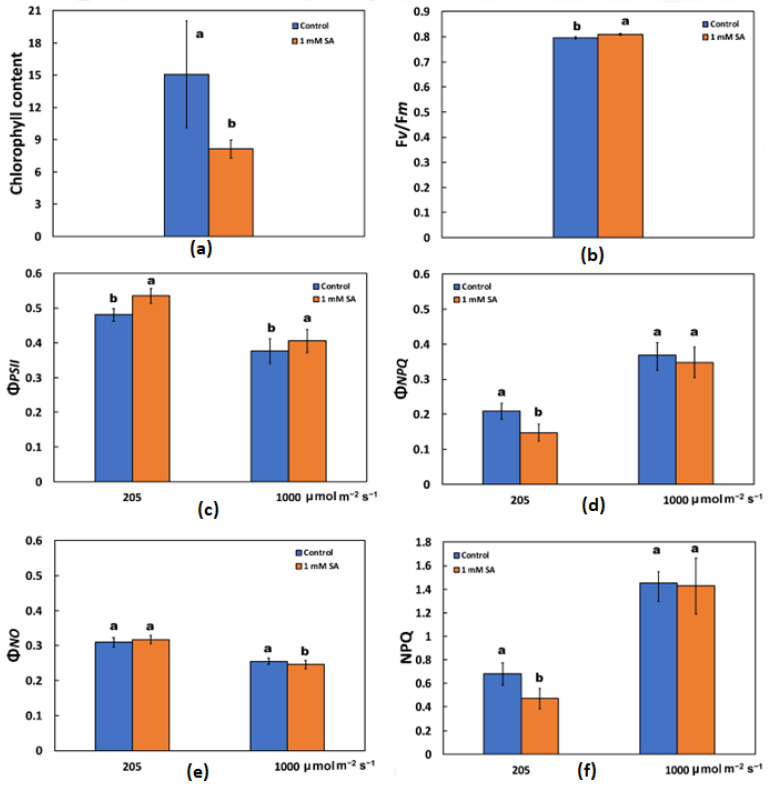
Chlorophyll contents, expressed in relative units (**a**); maximum efficiency of PSII photochemistry (F*v*/F*m*) (**b**); effective quantum yield of PSII photochemistry (Φ*_PSII_*) estimated at 205 and 1000 μmol photons m^−2^ s^−1^ actinic light (AL) intensity (**c**); quantum yield of regulated non-photochemical energy loss in PSII (Φ*_NPQ_*) estimated at 205 and 1000 μmol photons m^−2^ s^−1^ AL intensity (**d**); quantum yield of non-regulated energy dissipated in PSII (Φ*_NO_*) estimated at 205 and 1000 μmol photons m^−2^ s^−1^ AL intensity (**e**); and the non-photochemical quenching that reflects heat dissipation of excitation energy (NPQ) estimated at 205 and 1000 μmol photons m^−2^ s^−1^ AL intensity (**f**), in tomato leaves, after 72 h of foliar spray by either distilled water (control) or 1 mM salicylic acid (SA). Error bars are standard deviations (*n* = 4). Columns with different lowercase letters are statistically different (*p* < 0.05).

**Figure 2 ijms-23-07038-f002:**
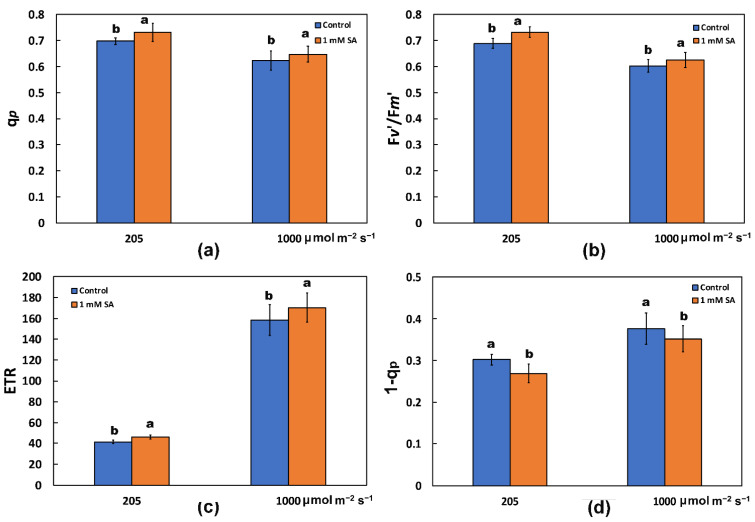
The fraction of open PSII reaction centers (q*p*), a measure of the redox state of quinone A (Q_A_) (**a**); the efficiency of excitation energy capture by open photosystem II reaction centers (F*v*’/F*m*’) (**b**); the electron transport rate (ETR) (**c**); and the excitation pressure (1 − q*p*) (data from a) (**d**); estimated at 205 and 1000 μmol photons m^−2^ s^−1^ actinic light (AL) intensity in tomato leaves, after 72 h of foliar spray by either distilled water (control) or 1 mM salicylic acid (SA). Error bars are standard deviations (*n* = 4). Columns with different lowercase letters are statistically different (*p* < 0.05).

**Figure 3 ijms-23-07038-f003:**
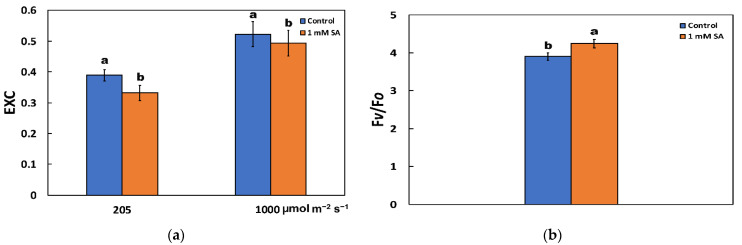
The relative excess energy at PSII (EXC), estimated at 205 and 1000 μmol photons m^−2^ s^−1^ actinic light (AL) intensity (**a**); and the efficiency of the oxygen evolving complex (OEC) on the donor side of PSII (F*v*/F*o*) (**b**); in tomato leaves after 72 h of foliar spray by either distilled water (control) or 1 mM salicylic acid (SA). Error bars are standard deviations (*n* = 4). Columns with different lowercase letters are statistically different (*p* < 0.05).

**Figure 4 ijms-23-07038-f004:**
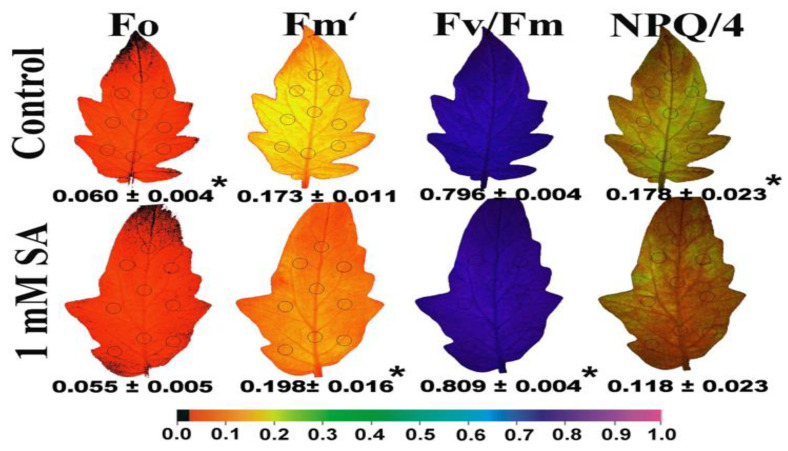
Representative color-coded images of the minimum chlorophyll *a* fluorescence in the dark (F*o*), the maximum chlorophyll *a* fluorescence in the light (F*m*‘), the maximum efficiency of PSII photochemistry (F*v*/F*m*), and the non-photochemical quenching (NPQ), depicted as NPQ/4, of tomato leaflets, after 72 h of foliar spray by either distilled water (control) or 1 mM salicylic acid (SA). The areas of interest (AOI) are shown in circles and the resulting values for each chlorophyll fluorescence parameter for the whole leaflet (average ± SD) are given for the dark adaptive state or under 205 μmol photons m^−^^2^ s^−^^1^ corresponding to low light intensity (LL). The color code showed at the bottom ranges from values 0.0 to 1.0. An asterisk indicates statistically significant difference (*p* < 0.05) between the control and 1 mM salicylic acid (SA) treatment.

**Figure 5 ijms-23-07038-f005:**
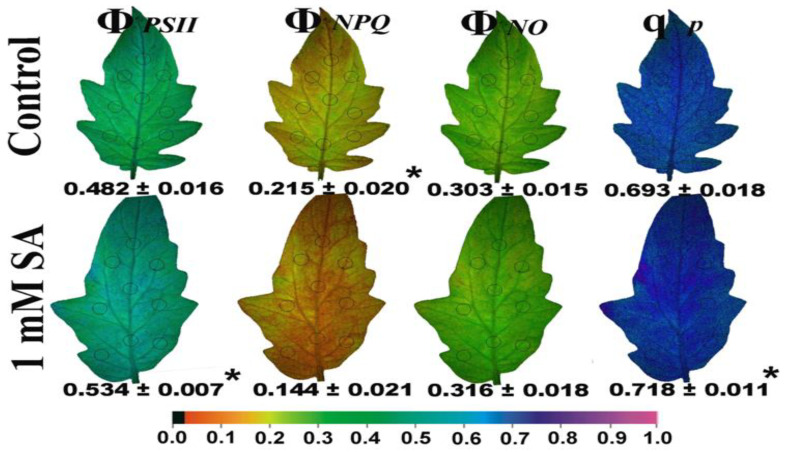
Representative color-coded images under actinic light (AL) of 205 μmol photons m^−^^2^ s^−^^1^, (LL), of the effective quantum yield of PSII photochemistry (Φ*_PSΙΙ_*), the quantum yield of regulated non-photochemical energy loss in PSII (Φ*_NPQ_*), the quantum yield of non-regulated energy loss in PSII (Φ*_NO_*), and the fraction of open PSII reaction centers (q*p*) of tomato leaflets after 72 h of foliar spray by either distilled water (control) or 1 mM salicylic acid (SA). The areas of interest (AOI) are shown in circles with values for each chlorophyll fluorescence parameter for the whole leaflet (average ± SD) are given. The color code showed at the bottom ranges from values 0.0 to 1.0. An asterisk indicates statistically significant difference (*p* < 0.05).

**Figure 6 ijms-23-07038-f006:**
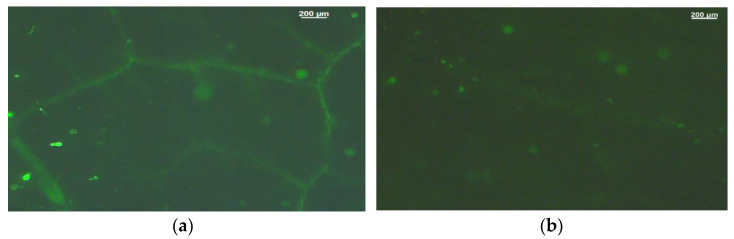
The ROS-specific fluorescence imaging of tomato leaflets stained by 2′, 7′-dichlorofluorescein diacetate (DCF-DA) after 72 h of foliar spray by either distilled water (control) (**a**); or 1 mM salicylic acid (SA) (**b**). Increased generation of ROS is visible by light green color. Scale Bar: 200 μm.

## Data Availability

Raw data used for this manuscript are available upon request to the corresponding author.

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
