# Peer review of "Harnessing the Role of Foliar Applied Salicylic Acid in Decreasing Chlorophyll Content to Reassess Photosystem II Photoprotection in Crop Plants"

_ijms, 2022, doi:10.3390/ijms23137038_

Round 1
Reviewer 1 Report
The manuscript of Moustakas et al. is a resubmission of a previous manuscript, which was reviewed by me pointing out a couple of statements, which were either too speculative and/or not sufficiently supported by the presented data. Some of these points were addressed in the revised and resubmitted manuscript. However, some points remained, which need further attention.
Problems:
Statistical treatment of the data
The effect of SA is very small in the case of several parameters, and the error bars are strongly overlapping for such parameters, which are indicated in the figures as being statistically significantly different. Based on my request the authors provided a table (as a non-publishable material) with the measured parameters and their statistical treatment. The mean values obtained from this table are indeed statistically different as shown in the figures. However, the supporting table contains only 28 parallel measurements for the untreated controls, and 30 parallel data series for the SA treatments instead of the expected 36 (which is calculated from the used 4 individual leaves with 9 points of interest for fluorescence detection in each of them). This points to the possibility that some of the data were neglected, raising concerns about the validity of the statistical significance of the complete dataset.
1O2 detection
The main hypothesis of the manuscript is that the application of SA decreases the Chl content (which is actually measured), and this effect is decreases the production of harmful 1O2, assuming that less Chl produces less triplet Chl, which in turn produces less 1O2. Unfortunately 1O2 was not measured directly it was only estimated from the efficiency on non-regulated fluorescence quenching. In the literature there are indeed theoretical work (e.g. by the cited Kasajima et al. paper), which predict that the efficiency of triplet Chl formation is proportional with the quantum yield of non-regulated fluorescence quenching. However, this prediction has not been experimentally verified, the only supporting indication is a modified expression level of genes, which are expected to be under 1O2-mediated control (the cited work Gawronski et al.). Therefore, the approach that minor changes in the quantum yield of non-regulated fluorescence quenching (4 % in Fig. 1E) could have any significant role in 1O2 production is not convincing.
Since the authors used a fluorescence sensor, DCF-DA, for H2O2 imaging it is quite surprising why did not apply the same approach for 1O2. Singlet Oxygen Sensor Green (SOSG) is a commercially available fluorescent 1O2 sensor, which is working reliably well in plants. It would have been a more useful approach to use SOSG and DCF-DA imaging in parallel to assess both 1O2 and H2O2 formation rather than to speculate on the 1O2 changes on the basis of non-regulated quenching.
The argumentation that the detected H2O2 is produced from 1O2 is also not too convincing. Although 1O2 can indeed induce other ROS (O2- and from that H2O2) it is certainly not the main pathway, therefore the conclusion that less Chl makes less 1O2 and that is responsible for the smaller level of detected H2O2 is unlikely.
Abstract
The purpose of the abstract is the present in a short form of new findings and conclusions from the work. Since the authors did not measure directly either the amount of triplet Chl or 1O2 the statement in the Abstract that “SA induced decrease in chlorophyll content resulted to
a lower amount of triplet excited state chlorophyll (3Chl*) molecules available to produce singlet oxygen (1O2) under high light treatment” is not a new finding just a suggestion, which is based on theoretical considerations from the literature. This sentence should be replaced by something which says, that SA treatment decreased non-regulated fluorescence quenching which indicate that less 3Chl and consequently less 1O2 could be formed.
Suggested changes:
In order to convince the readers that this is a meaningful work the following changes should be made:
1, The issue with the statistical treatment of the data should be solved. The obvious thing to do is to provide the raw data together with their statistical treatment as a PUBLISHED supporting material. This is an absolute requirement for most of the (better) journals. The table should indicate which points belong to each of the 4 leaves instead of the continuous listing of the 28-30 datalines.
2, It should be made clear that 1O2 estimation was based on the theoretically deduced proportionality between the quantum yield of non-regulated fluorescence quenching and 3Chl formation, which has not been experimentally verified.
3, The Abstract should be modified as described above.
Author Response
The manuscript of Moustakas et al. is a resubmission of a previous manuscript, which was reviewed by me pointing out a couple of statements, which were either too speculative and/or not sufficiently supported by the presented data. Some of these points were addressed in the revised and resubmitted manuscript. However, some points remained, which need further attention.
Problems:
Statistical treatment of the data
The effect of SA is very small in the case of several parameters, and the error bars are strongly overlapping for such parameters, which are indicated in the figures as being statistically significantly different. Based on my request the authors provided a table (as a non-publishable material) with the measured parameters and their statistical treatment. The mean values obtained from this table are indeed statistically different as shown in the figures. However, the supporting table contains only 28 parallel measurements for the untreated controls, and 30 parallel data series for the SA treatments instead of the expected 36 (which is calculated from the used 4 individual leaves with 9 points of interest for fluorescence detection in each of them). This points to the possibility that some of the data were neglected, raising concerns about the validity of the statistical significance of the complete dataset.
We have provided you with the raw data of the chlorophyll fluorescence measurements and you checked that the mean values obtained from this table are indeed statistically different despite the fact that the error bars are strongly overlapping. After checking them yourself, stating neglection of the raw data is a serious accusation suggesting misconduct of science in our behalf.
The difference in the number of values is due to the fact that in some leaves there were chosen 6 areas of interest (AOI) and in some others 7,8, or 9, depending on the leaf morphology and leaf area. We have inserted in the text (lines 435-436) this detail.
1O2 detection
The main hypothesis of the manuscript is that the application of SA decreases the Chl content (which is actually measured), and this effect is decreases the production of harmful 1O2, assuming that less Chl produces less triplet Chl, which in turn produces less 1O2. Unfortunately 1O2 was not measured directly it was only estimated from the efficiency on non-regulated fluorescence quenching. In the literature there are indeed theoretical work (e.g. by the cited Kasajima et al. paper), which predict that the efficiency of triplet Chl formation is proportional with the quantum yield of non-regulated fluorescence quenching. However, this prediction has not been experimentally verified, the only supporting indication is a modified expression level of genes, which are expected to be under 1O2-mediated control (the cited work Gawronski et al.). Therefore, the approach that minor changes in the quantum yield of non-regulated fluorescence quenching (4 % in Fig. 1E) could have any significant role in 1O2 production is not convincing.
We have changed the relative sentence in Abstract as you suggested.
Since the authors used a fluorescence sensor, DCF-DA, for H2O2 imaging it is quite surprising why did not apply the same approach for 1O2. Singlet Oxygen Sensor Green (SOSG) is a commercially available fluorescent 1O2 sensor, which is working reliably well in plants. It would have been a more useful approach to use SOSG and DCF-DA imaging in parallel to assess both 1O2 and H2O2 formation rather than to speculate on the 1O2 changes on the basis of non-regulated quenching.
We will have in mind to use in our future works Singlet Oxygen Sensor Green (SOSG).
The argumentation that the detected H2O2 is produced from 1O2 is also not too convincing. Although 1O2 can indeed induce other ROS (O2- and from that H2O2) it is certainly not the main pathway, therefore the conclusion that less Chl makes less 1O2 and that is responsible for the smaller level of detected H2O2 is unlikely.
We clearly state (lines 380-381) that: “However, electron leakage to O2 at PSI, that results to O2•–, which is converted to H2O2, is the main pathway of H2O2 generation [39,44,45].”
Abstract
The purpose of the abstract is the present in a short form of new findings and conclusions from the work. Since the authors did not measure directly either the amount of triplet Chl or 1O2 the statement in the Abstract that “SA induced decrease in chlorophyll content resulted to a lower amount of triplet excited state chlorophyll (3Chl*) molecules available to produce singlet oxygen (1O2) under high light treatment” is not a new finding just a suggestion, which is based on theoretical considerations from the literature. This sentence should be replaced by something which says, that SA treatment decreased non-regulated fluorescence quenching which indicate that less 3Chl and consequently less 1O2 could be formed.
The Abstract was modified as you suggested.
Suggested changes:
In order to convince the readers that this is a meaningful work the following changes should be made:
1, The issue with the statistical treatment of the data should be solved. The obvious thing to do is to provide the raw data together with their statistical treatment as a PUBLISHED supporting material. This is an absolute requirement for most of the (better) journals. The table should indicate which points belong to each of the 4 leaves instead of the continuous listing of the 28-30 data lines.
We have provided you with the raw data of the chlorophyll fluorescence measurements and you checked that the mean values obtained from this table are indeed statistically different despite the fact that the error bars are strongly overlapping. We also could not believe that there was a significant difference before conducting the statistical analysis. The difference in the number of values is due to the fact that in some leaves there were chosen 6 areas of interest (AOI) and in some others 7,8 or 9. We have inserted in the text (lines 435-436) this detail. Providing the raw data in tables is not a common practice in this type of measurements therefore we do not think we should provide them.
If you can name 5 articles that have raw data of chlorophyll fluorescence measurements in “better” journals then please let us know.
However, we solved what you name “the issue with the statistical treatment of the data” by including on line 501:
Data Availability Statement: Raw data used for this manuscript are available upon request to the corresponding author.
2, It should be made clear that 1O2 estimation was based on the theoretically deduced proportionality between the quantum yield of non-regulated fluorescence quenching and 3Chl formation, which has not been experimentally verified.
This is clear both in Abstract (lines 30-33) and in Discussion (lines 369-378, 393-397)
3, The Abstract should be modified as described above.
The Abstract was modified as you suggested.
Reviewer 2 Report
My correction with the manuscript are relatively minor, and are listed below. I can recommend publication of this paper after these corrections. The topic of the study is interesting and it fits into trends in science as well as in the manufacturing practice. The title cover the main aspects in this paper, it reflect the aim and scientific purpose of conducted experiment. Presented abstract explain the significance of the paper and contain the background, results and conclusion parts. Introduction provides a good, generalized background of the topic giving the reader an appreciation of the wide range of applications for this technology. The results are clearly explained and presented in an appropriate format. Suggestion to the authors is to correct the legend on Figure 1: "1MM" to "1mM SA". The methods used in this paper are appropriate to the aim of the study. Methods are clear and replicable. Paper contain sufficient information for a capable researcher to reproduce the experiments described. Conclusions presented in this paper correlate to the results found. Given the scope of the results presented please describe research limitations for future research.
Author Response
My correction with the manuscript are relatively minor, and are listed below. I can recommend publication of this paper after these corrections. The topic of the study is interesting and it fits into trends in science as well as in the manufacturing practice. The title cover the main aspects in this paper, it reflect the aim and scientific purpose of conducted experiment. Presented abstract explain the significance of the paper and contain the background, results and conclusion parts. Introduction provides a good, generalized background of the topic giving the reader an appreciation of the wide range of applications for this technology. The results are clearly explained and presented in an appropriate format. Suggestion to the authors is to correct the legend on Figure 1: "1MM" to "1mM SA". The methods used in this paper are appropriate to the aim of the study. Methods are clear and replicable. Paper contain sufficient information for a capable researcher to reproduce the experiments described. Conclusions presented in this paper correlate to the results found. Given the scope of the results presented please describe research limitations for future research.
Thank you for your comments. In Figure 1 we changed "1MM" to "1mM SA". We have added also scope for future research (lines 411-414).
Reviewer 3 Report
I have reviewed this MS with a good deal of zeal and zest. I found the manuscript to be accepted for publication. I wonder if the authors could include a list of abbreviations used,
Moreover, if the author could add some regression analysis b/w chlorophyll contents and efficiency of photosystem II.?
How the low and high concentrations can affect the observed parameters, This fact should be discussed more in the discussion section.
Author Response
Ι have reviewed this MS with a good deal of zeal and zest. I found the manuscript to be accepted for publication. I wonder if the authors could include a list of abbreviations used
We included in Appendix A an abbreviation list.
Moreover, if the author could add some regression analysis b/w chlorophyll contents and efficiency of photosystem II.?
Since we have used regression analysis in a number of previous articles [Molecules 26:4157 (2021); Environmental and Experimental Botany 154: 44–55 (2018); Photosynthetica 53: 471-477 (2015); Journal of Plant Research 127: 481-489 (2014); Journal of Plant Physiology 171: 587-593 (2014); Journal of Plant Physiology 169: 577-585 (2012); Acta Physiologiae Plantarum 34: 1267-1276 (2012)] we did not want to apply the same approach in this manuscript.
How the low and high concentrations can affect the observed parameters, This fact should be discussed more in the discussion section.
We have used one concentration of SA with low and high light intensity measurements of the chlorophyll fluorescence parameters and when there are differences these are discussed in text e.g., ΦPSI (lines 146-147), ΦNPQ (lines 148-149, 366-369), ΦNO (lines 150-151, 369-371, 394-395).
This manuscript is a resubmission of an earlier submission. The following is a list of the peer review reports and author responses from that submission.
Round 1
Reviewer 1 Report
The manuscript of Moustakas et al. deals with the effect of foliar application of salicylic acid on the photosynthetic parameters of tomato plants. The authors have measured various parameters of photosynthetic activity by Chl fluorescence imaging, as well as “ROS” by DCF-DA imaging. They claim that foliar applied SA induced chlorophyll breakdown from thylakoid membranes lowering ROS generation and enhancing PSII efficiency and also that chlorophyll catabolism stimulated by SA treatment can be regarded as a mechanism that can lower photoinhibition and photodamage of PSII. Unfortunately, there are several problems with the work as outlined in the comments below, which makes unsuitable the manuscript for publication.
Problems:
1, The title says that “Salicylic Acid-Induced Chlorophyll Catabolism in Tomato Leaves Reduced Photoinhibition and Photodamage of Photosystem II by Suppressing Reactive Oxygen Species Creation”. Unfortunately none of the key features (i.e. Chl catabolism, photoinhibition, photodamage) were studied in the work. Chl content was measured, but nothing proves that it was catabolized. Were there any Chl breakdown products detected? There is an attempt to detect ROS production by DCF-DA imaging, which is also problematic (see below). Therefore, the title does not reflect the content of the manuscript.
If the work is intended to say something about photodinhibition or photodamage then these parameters should be directly measured, not only speculated about.
2, Lack of scientific novelty. Almost all findings of the work (i.e. the effect of SA on Chl content, photosynthetic activity parameters) have been measured and published before, in several cases resulting with opposite conclusions than those reported here. The only novel thing would be the ROS detection, but see its problems below. Therefore, the work lacks significant novelty regarding our knowledge about the effects of SA on photoinhibition or PSII photodamage.
3, Statistical analysis. The manuscript draws conclusions about statistically significant changes in cases when there is only a minor difference in the mean values and the error bars are largely overlapping (Fig. 1b, Fig. 2b 1000 uE, Fig. 3a 1000 uE, Fig. 4a and b 1000 uE, Fig. 5a and b, Fig. 6a 1000 uE). As a rule of thumb in case of low number of repetitions (4 in this work) significant differences can be expected only if the error bars are not overlapping. Of course there might be alterations from this approximate rule, which can be verified by statistical analysis. Since the above listed examples look questionable the authors should present all of the raw data from which the figures and statistical calculations were made in the form of a supplementary Excel table which can be used for recalculation of the statistical analysis. In the absence of such a table most of the conclusions are hanging in the air, since the figures do not support the conclusions.
4, Literature citations. The manuscript puts emphasis on the damaging role of ROS, especially singlet oxygen, on the function of PSII. Unfortunately none of the cited papers gives a clear view about the formation and elimination of 1O2 in the photosynthetic apparatus. Papers like Physiologia Plantarum 142: 6 – 16. 2011, Biochimica et Biophysica Acta 1827 (2013) 689–698 could help clarifying the background of the 1O2.
5, ROS detection. The authors used DCF-DA imaging for ROS detection. The problem with DCF-DA is that it is highly unselective and reacts mostly with H2O2, but not with 1O2. Therefore, it is unsuitable for detection of Chl-derived 1O2 production, which would be the aim of the present manuscript. Fig. 9. is anyway unsuitable for any conclusion. The bright green spots in the controls leaves (panel a) must come from contaminations. There are some hardly visible continuous green lines, which could be the contour of plant cells, but this does not make much sense either. Why would H2O2 be formed at the cell wall far away from the chloroplasts? In addition, this is a single measurement, the apparent difference between the control and the SA-treated leaves may come from the uncertainty of the method.
The main pathway for H2O2 production is certainly not via 1O2, therefore the conclusion in the Discussion (lines 368-375) that decreased Chl content would decrease H2O2, remains an empty speculation.
1O2 in plant leaves can be well detected by another fluorescent sensor SOSG (Singlet Oxygen Sensor Green, see Journal of Experimental Botany, Vol. 57, No. 8, pp. 1725–1734, 2006). In order to support any conclusion about the effect of SA on 1O2 production the SOSG imaging method should be used instead of the DCF-DA imaging.
6, ETR determination. The ETR value shown in Fig. 5a corresponds to ETR of PSII, which should be specified since there is an ETR of PSI as well, which is different from ETR(II).
7, Generalization. The concluding sentences (lines 302-306) that refer to an earlier result of SA on tomato plants, which is just the opposite to that reported here decrease further the trust in the validity of the results (which is already undermined by the problems with the statistical significance, point 3.) : “Salicylic acid (1 mM), supplied in the nutrient solution of tomato plants for 24 h, decreased the maximum efficiency of PSII (Fv/Fm), and the effective quantum yield of PSII(ΦPSII) [78]. It seems that most results obtained with exogenous SA cannot be generalized, since the effect may differ not only with plant species, but it may vary on the method of SA application (spraying, growth medium addition, seed pre-soaking, etc.), as well as on the experimental time.” This conclusion shows that we learn nothing about the function of SA if it has so contradicting effects depending on the application method ans other experimental conditions.
Author Response
1, The title says that “Salicylic Acid-Induced Chlorophyll Catabolism in Tomato Leaves Reduced Photoinhibition and Photodamage of Photosystem II by Suppressing Reactive Oxygen Species Creation”. Unfortunately none of the key features (i.e. Chl catabolism, photoinhibition, photodamage) were studied in the work. Chl content was measured, but nothing proves that it was catabolized. Were there any Chl breakdown products detected? There is an attempt to detect ROS production by DCF-DA imaging, which is also problematic (see below). Therefore, the title does not reflect the content of the manuscript.
The title and the text have changed taking into account the comments.
If the work is intended to say something about photodinhibition or photodamage then these parameters should be directly measured, not only speculated about.
Photoinhibition and photodamage have been directly measured and are not speculated.
In Introduction (L. 76-81) we refer to photoinhibition and how it is measured “Chlorophyll fluorescence is widely used to measure photoinhibition, based on the ratio Fv/Fm, the maximum efficiency of PSII photochemistry [12,14].” We used Fv/Fm measurements (Figure 1b).
Photodamage (L. 264-269) results in ROS production. We have measured ROS by DCF-DA imaging which it is a widely used method. Still, the consequences of photodamage through the triplet excited state chlorophylls (3Chl*) is the generation of singlet oxygen (1O2). We have assessed 1O2 generation through ΦNO measurements.
2, Lack of scientific novelty. Almost all findings of the work (i.e. the effect of SA on Chl content, photosynthetic activity parameters) have been measured and published before, in several cases resulting with opposite conclusions than those reported here. The only novel thing would be the ROS detection, but see its problems below. Therefore, the work lacks significant novelty regarding our knowledge about the effects of SA on photoinhibition or PSII photodamage.
We are not aware of another article that has reached out in the same conclusions.
3, Statistical analysis. The manuscript draws conclusions about statistically significant changes in cases when there is only a minor difference in the mean values and the error bars are largely overlapping (Fig. 1b, Fig. 2b 1000 uE, Fig. 3a 1000 uE, Fig. 4a and b 1000 uE, Fig. 5a and b, Fig. 6a 1000 uE). As a rule of thumb in case of low number of repetitions (4 in this work) significant differences can be expected only if the error bars are not overlapping. Of course there might be alterations from this approximate rule, which can be verified by statistical analysis. Since the above listed examples look questionable the authors should present all of the raw data from which the figures and statistical calculations were made in the form of a supplementary Excel table which can be used for recalculation of the statistical analysis. In the absence of such a table most of the conclusions are hanging in the air, since the figures do not support the conclusions.
You can find attached the statistical analysis for your own use in a word document. We cannot upload in the system xls files, so we cannot upload raw data. We can send them by e-mail personally. We can also confirm that we have not estimated significance by eye.
Concerning your comment on the number of repetitions for chlorophyll fluorescence data the average number of samples used in chlorophyll fluorescence studies in the lab is 3 to 6. You can verify this in the list of articles that follows (this list can be extended if needed).
Plant Cell Physiol. 2011, 52, 1822–1831. (Fig. 3, n= 3). (Fig. 4, n= 3).
J Plant Physiol. 2014, 171, 23-30. (Four replicates, each consisting of an individual plant in a pot)
Plant Cell Physiol. 2016, 57, 1510-1517. (Three independent measurements)
Physiol. Plant. 2016, 158, 225–235. (Four biological replicates)
Front. Plant Sci. 2016, 7, 453 (4 replicates)
Science 2016, 354 (6314) 857-860. (Fig. 3, n = 5 biological replicates), (Fig. 4, n = 6 biological replicates)
Sci. Rep. 2017, 7, 46100. https://doi.org/10.1038/srep46100. (n= 3).
Nature Plants 2017, 3, 17033. DOI: 10.1038/nplants.2017.33 (Figure 3. n > 3).
ACS Appl. Mater. Interfaces 2018, 10, 4450−4461 (n= 5).
Plant Cell Environ. 2021, 44, 3002–3014. (n = 3, biologically independent samples)
J Plant Physiol. 2021, 262, 153438 (N=3, N=4)
J Plant Physiol. 2021, 260, 153404 (three replicates)
Journal of Hazardous Materials 2021, 404, 124001 (three independent biological replicates)
Molecules 2021, 26, 2984. https://doi.org/10.3390/molecules26102984 (four independent measurements)
Int. J. Mol. Sci. 2021, 22, 41. https://dx.doi.org/10.3390/ijms22010041. (n = 5).
4, Literature citations. The manuscript puts emphasis on the damaging role of ROS, especially singlet oxygen, on the function of PSII. Unfortunately none of the cited papers gives a clear view about the formation and elimination of 1O2 in the photosynthetic apparatus. Papers like Physiologia Plantarum 142: 6 – 16. 2011, Biochimica et Biophysica Acta 1827 (2013) 689–698 could help clarifying the background of the 1O2.
We included in our Discussion the paper Physiologia Plantarum 142: 6 – 16 (2011). In many parts of the manuscript (L.112-118, 264-266, 358-375) a clear view of 1O2 formation and elimination is given.
5, ROS detection. The authors used DCF-DA imaging for ROS detection. The problem with DCF-DA is that it is highly unselective and reacts mostly with H2O2, but not with 1O2. Therefore, it is unsuitable for detection of Chl-derived 1O2 production, which would be the aim of the present manuscript. Fig. 9. is anyway unsuitable for any conclusion. The bright green spots in the controls leaves (panel a) must come from contaminations. There are some hardly visible continuous green lines, which could be the contour of plant cells, but this does not make much sense either. Why would H2O2 be formed at the cell wall far away from the chloroplasts? In addition, this is a single measurement, the apparent difference between the control and the SA-treated leaves may come from the uncertainty of the method.
We have not used DCF-DA imaging for 1O2 detection. There is no mention on the text that we used DCF-DA to probe 1O2. We agree that DCF-DA is used as an unselective ROS probe, and reacts mostly with H2O2. We used DCF-DA for probing ROS in general. Singlet oxygen (1O2) production was assessed by ΦNO measurements.
There is no mention also about H2O2 to be formed at the cell wall. In lines 236-237 we wrote “A higher ROS generation was visible in control leaves as green fluorescence localized mainly in leaf veins and leaf hairs (Figure 6a)”. Still, we do not present results after one measurement.
H2O2 is the most stable and least reactive ROS with the longest lifetime being able to easily diffuse through the membranes [J. Exp. Bot. 2021, 72, 5857-5875].
The main pathway for H2O2 production is certainly not via 1O2, therefore the conclusion in the Discussion (lines 368-375) that decreased Chl content would decrease H2O2, remains an empty speculation.
Yes, the main pathway for H2O2 is certainly not via 1O2 and we do not claim this. The Discussion (lines now 367-376) does not refer to any decrease of H2O2. We do not claim what it is written in your comment. We wrote in Introduction (lines 115-117) “Electron leakage to O2 at PSI results to the superoxide anion radical (O2•–), that via a disproportionation reaction catalysed by superoxide dismutase (SOD) is converted to hydrogen peroxide (H2O2) [44,45]”. And to clarify the main pathway for H2O2 in Discussion we added (lines 371-372) “However, the main pathway of H2O2 generation is at PSI though electron leakage to O2 that results to O2•–, that is converted to H2O2 [39,44,45]”.
1O2 in plant leaves can be well detected by another fluorescent sensor SOSG (Singlet Oxygen Sensor Green, see Journal of Experimental Botany, Vol. 57, No. 8, pp. 1725–1734, 2006). In order to support any conclusion about the effect of SA on 1O2 production the SOSG imaging method should be used instead of the DCF-DA imaging.
We have detected ROS in general by DCF-DA imaging and we did not use DCF-DA to probe 1O2 production. Singlet oxygen (1O2) production was assessed by ΦNO measurements.
6, ETR determination. The ETR value shown in Fig. 5a corresponds to ETR of PSII, which should be specified since there is an ETR of PSI as well, which is different from ETR(II).
We have measured only PSII photochemistry so there is no need to specify which ETR it is.
7, Generalization. The concluding sentences (lines 302-306) that refer to an earlier result of SA on tomato plants, which is just the opposite to that reported here decrease further the trust in the validity of the results (which is already undermined by the problems with the statistical significance, point 3.) : “Salicylic acid (1 mM), supplied in the nutrient solution of tomato plants for 24 h, decreased the maximum efficiency of PSII (Fv/Fm), and the effective quantum yield of PSII(ΦPSII) [78]. It seems that most results obtained with exogenous SA cannot be generalized, since the effect may differ not only with plant species, but it may vary on the method of SA application (spraying, growth medium addition, seed pre-soaking, etc.), as well as on the experimental time.” This conclusion shows that we learn nothing about the function of SA if it has so contradicting effects depending on the application method ans other experimental conditions.
What you state about the statistical significance of our data and the validity of our results based on the previous opposite results on tomato, are not documented. In contrast to your statement, the previous opposite results on tomato increase the validity of our data.
In the previous experiment salicylic acid (1 mM), was supplied in the nutrient solution (lines 308-309). In our work we applied SA by spray in the leaves. Still, (lines 461-464) “According to our experimental data, foliar application of 1 mM SA by spray, compared to other application methods, has the best beneficial effect by alleviating photoinhibition and photodamage and improving PSII efficiency in the crop plant tomato.”
The conclusion you refer to your comment shows exactly the opposite from your statement. It shows that we learned that the effect of SA on plants depends on numerous aspects (lines 457-461) “the impact of SA on plant growth and physiology relies highly on the study design strategy and depends significantly on numerous aspects, such as the plant species or the genotype, the environmental conditions, the concentration used, the duration of exposure, and mainly the method of application (foliar, rooting medium e.g., soil or hydroponic).”

Reviewer 2 Report
The manuscript entitled “Salicylic Acid-Induced Chlorophyll Catabolism in Tomato Leaves Reduced Photoinhibition and Photodamage of Photo-system II by Suppressing Reactive Oxygen Species Creation” seems to fit the aims of the scientific journal IJMS. This manuscript examines the consequences of foliar application of 1 mM SA for 72 h on photosystem II (PSII) function and investigates the effect of SA on reactive oxygen species (ROS) and chlorophyll content to elucidate the molecular mechanism of the effect of SA on electron transport. which is still unclear under stress-free conditions. The results suggest that foliar application of SA induces the degradation of chlorophyll from thylakoid membranes, which decreases the formation of ROS and increases PSII efficiency. Consequently, this study sheds light on the role that chlorophyll molecules play in regulating PSII efficiency as well as ROS generation. The topic of the study is interesting and it fits into trends in science as well as in the manufacturing practice.
- The title cover the main aspects in this paper, it reflect the aim and scientific purpose of conducted experiment. Presented abstract explain the significance of the paper and contain the background, results and conclusion parts.
- Introduction provides a good, generalized background of the topic giving the reader an appreciation of the wide range of applications for this technology.
- The results are clearly explained and presented in an appropriate format. Suggestion to the authors is to correct the legend on Figure 1: "1MM" to "1mM SA".
- The methods used in this paper are appropriate to the aim of the study. Methods are clear and replicable. Paper contain sufficient information for a capable researcher to reproduce the experiments described.
- Conclusions presented in this paper correlate to the results found. Given the scope of the results presented, it is necessary to improve conclusion section. Please rewrite it clearly stating the facts; focus more on how your research has contributed to knowledge gaps; describe research limitations for future research and restate your major findings; add scientific and practical significance of the selected method.
My correction with the manuscript are relatively minor, and are listed below. I can recommend publication of this paper after these corrections.
Author Response
- The title cover the main aspects in this paper, it reflect the aim and scientific purpose of conducted experiment. Presented abstract explain the significance of the paper and contain the background, results and conclusion parts.
- Introduction provides a good, generalized background of the topic giving the reader an appreciation of the wide range of applications for this technology.
- The results are clearly explained and presented in an appropriate format. Suggestion to the authors is to correct the legend on Figure 1: "1MM" to "1mM SA".
It was corrected
- The methods used in this paper are appropriate to the aim of the study. Methods are clear and replicable. Paper contain sufficient information for a capable researcher to reproduce the experiments described.
- Conclusions presented in this paper correlate to the results found. Given the scope of the results presented, it is necessary to improve conclusion section. Please rewrite it clearly stating the facts; focus more on how your research has contributed to knowledge gaps; describe research limitations for future research and restate your major findings; add scientific and practical significance of the selected method.
We rewrote conclusion section as you suggested.
Reviewer 3 Report
The authors state SA as oxidative growth regulatoors which is a poor word choice, rather redundant.
Whole abstract is rather generic and can be modified for better quantitative perspectives.
While, the whole wordy introduction covers the story, there should be a gap statement. A well developed hypothesis and how this study is different among many others in the field.
The results section is ok, however the figure quality must be improved.
Materials and methods, ok
Discussion section is upto date and necesssary interpretations are included within it. However, the refernces can be enriched with more recent citations.
Conclusion is fine and supported by obtained results.
Author Response
The authors state SA as oxidative growth regulatoors which is a poor word choice, rather redundant.
Yes, you are right. We change it to “acting as an antioxidant and a plant growth regulator”. Thank you for pointing it.
Whole abstract is rather generic and can be modified for better quantitative perspectives.
Minor modifications were made in the Abstract.
While, the whole wordy introduction covers the story, there should be a gap statement. A well developed hypothesis and how this study is different among many others in the field.
We have formulated a new hypothesis taking into account your suggestion. Still the Title has been changed.
The results section is ok, however the figure quality must be improved.
We improved the figure quality.
Materials and methods, ok
Discussion section is upto date and necesssary interpretations are included within it. However, the refernces can be enriched with more recent citations.
We could have enriched the refernces with more recent citations but the number of citations is already high (114).
Conclusion is fine and supported by obtained results.
Round 2
Reviewer 1 Report
The manuscript of Moustakas was somewhat improved as a result of the revision, but still has critical problems and unsuitable for publication in the present form. The most critical issues are: (i) Lack of confidence in the statistical significance of the data, and (ii) Inadequate detection of 1O2 via ΦNO measurements. In order to deal with the firs point the authors should provide the raw data as a supplementary material in a table format (Word, PDF, whatever can be uploaded) for Figures 1, 2 and 3. In order to deal with the second point the whole discussion about SA effect on Chl-derived 1O2 production should be toned down since there is no experimental proof for it.
Insufficient statistical treatment of the data
As I pointed out in the original review the error bars are overlapping to such a large extent, which makes highly questionable that the indicated differences could be statistically significant. In the present form the data simply do not support the conclusions. The authors refused my simple request to provide the original data as supplementary information, which is a minimum requirement for many scientific journals. I did not receive any statistical calculation either. This is not required only to persuade the reviewer, but also the readers, since by looking at the figures the (small differences with largely overlapping error bars) the first thought is that something is “fishy” here. The supplementary data do not have to be in Excel format, Word or PDF is also good. (Anyway, to refuse the presentation of raw data is not too fortunate, since that could easily clear the situation.) In the respect of statistical significance the number of repetitions is not critical 4-6 independent measurements are fine, the problem is with the large, overlapping error (which makes questionable the claimed statistical significance).
There is no experimental evidence for that 1O2 production could be measured by ΦNO.
One of the critical point is the determination of the rate (amount) of singlet oxygen. As I pointed out in my original review the imaging by the “ROS” sensor DCF-DA is unsuitable for that since it does not react with 1O2. The authors’ answer to this point is that they used the value of the yield of non-regulated fluorescence quenching, ΦNO, to determine the amount of 1O2. They cite the following papers as a support for this method. Most of these papers are published by the authors (of this manuscript) without any experimental proof that ΦNO and 1O2 production would be directly correlated. Other papers either do not mention ΦNO and 1O2, or their correlation, or cite back to earlier works. Ref. 41 states that the non-regulated process consists of chlorophyll fluorescence, internal conversion, and intersystem crossing. However, this is still the sum of different processes, which can change independently of each other (chlorophyll fluorescence is certainly not involved in singlet oxygen formation). Therefore, there might be an indirect non-linear correlation between ΦNO and 1O2 production, ΦNO in itself is certainly not a measure of 1O2.
Among the cited papers there is only one (Ref. 95), which shows that under conditions of
increased ΦNO the expression of some genes, which are expected to be expressed by 1O2 are
increased. However, this result is not sufficient to support the idea that 1O2 production could
be measured by ΦNO.
The following papers were cited for the putative correlation of ΦNO and 1O2 production:
Ref. 40 (Müller et al.): No mention of F(NO).
Ref. 41 (Kasajima et al.) Gives the following definition of non-regulated dissipation: „(NO) Basal dissipation/ non-light induced quenching consists of chlorophyll fluorescence, internal conversion, and intersystem crossing”, which means that the NO process involves not only the potentially 1O2 producing intersystem crossing. This paper does not deal with 1O2 formation, or its connection with F(NO).
Ref. 42 (Moustaka et al.) Own paper of the authors, cites back to earlier papers, no direct experimental support for the correlation of ΦNO and 1O2 formation.
Ref. 43 (Sperdouli et al.) Own paper of the authors, cites back to earlier papers, no direct experimental support for the correlation of ΦNO and 1O2 formation.
Ref. 64 (Vass) Does not mention ΦNO and its putative correlation with 1O2 formation.
Ref. 94 (Prasad et al.) Does not mention ΦNO, and/or its putative correlation with 1O2 formation.
Ref. 95 (Gawronksi et al.) Measured ΦNO, and the expression of some genes, which are expected to be upregulated by 1O2, but no direct relationship between ΦNO and 1O2 formation was demonstrated. Klughammer and Schreiber (2008) is not cited here, but in some of the above papers. This gives a clear theoretical background for the non-regulated dissipation process and speculates about its possible correlation with 1O2 production, but without experimental support.
The issue of 1O2 production
As pointed out above there is no experimental evidence for the direct correlation of 1O2 production and ΦNO. As a consequence of a simple physical law the rate of 1O2 production from antenna Chl-s is proportional with light intensity (higher excitation makes more 3Chl, which makes more 1O2). Therefore, a ca. 5-times more 1O2 (5-fold higher rate) is expected at 1000 E as compared to 205 E. However, Fig. 1 E, shows that ΦNO is not 5-fold higher, but smaller at 1000 E than at 205 E. This again does not make sense with the idea that the value of would reflect 1O2 production. Anyway, according the Fig. 1E the value ΦNO is maximum 2-3% smaller in the presence of SA than in the control (if indeed significant). This is such a minor effect that cannot cause any significant photoprotection.
DCF-DA imaging
Why would any ROS species, which are generated from Chl accumulate in leaf veins and leaf hairs? This does not make much sense.
The question of the application method of SA
I accept that the effect of SA depends on the way of application. However, it is still highly problematic that the effect of SA is just the opposite when it is applied via foliar spraying and uptake from roots. This indicates something that the SA effect is highly indirect and very little is known about its molecular background.